# External Debt Determinants: Do Macroeconomic and Institutional Ones Matter for Selected ASEAN Developing Countries?

**Edi Harsono [1,*]**, **Andi Kusumawati [2,*]** and **Nirwana Nirwana [2]**

[1] Directorate of General of Taxes, Ministry of Finance, Jakarta 10110, Indonesia
[2] Departement of Accounting, Faculty of Economics and Business, Hasanuddin University, Makassar 90245, Indonesia
[*] Correspondence: ediharsono@kemenkeu.go.id (E.H.); andikusumawati@unhas.ac.id (A.K.)

**Abstract:** Developing nations have the task of effectively managing their external debt. The government is urged to comprehend the decisive component in managing its external debt, despite the varying viewpoints among economists. In addition, the world sees the need for institutional quality to optimize its economic policy. Institutional quality shows accountability, stability, effectiveness, quality, law, and trust. Our research examines the determinant factors of external debt and discusses the policy to manage external debt. We regress the inflation rate, exchange rate, interest rate, trade openness, and institutional quality on external debt. This study also uses moderated regression analysis to examine the interaction between institutional quality and macroeconomic indicators on external debt. We selected 52 samples from five ASEAN developing countries from 2008 to 2019. The first study found that the inflation rate, interest rate, and institutional quality have a negative impact on external debt, while the exchange rate and trade openness have a positive impact on external debt. Next, we were surprised that institutional quality could not moderate the relationship between the inflation rate, exchange rate, and interest rate on external debt. Further, it only moderated the relationship between trade openness and external debt. In the end, we discuss the external debt determinants from the selected ASEAN developing countries with the theories.

**Keywords:** external debt; macroeconomic indicator; institutional quality; inflation rate; exchange rate; interest rate; trade openness

## 1. Introduction

The external debt of many countries has increased over the decades. Annual accumulation is a common developing country characteristic in the early stage of economic development (Martin 2009; Beyene and Kotosz 2020). Lau et al. (2022) stated that external debt is one of the important resources for growth in Asian developing countries. Debt provides fresh funds for the government to cover its fiscal budget shortfalls. Then, the government stimulates consumption in the household sector, and economic growth is created.

Based on the World Bank database, Figure 1 shows how Southeast Asian developing countries experienced managing external debt and achieved their gross domestic product (GDP) for 2012–2021. The following image displays a graphical bar and colored maps processed from developing countries that are members of the Association of Southeast Asian Nations (ASEAN). The blue bar shows the amount of the external debt, and the red bar displays GDP. From the data displayed, there has been an increase in the amount of external debt in all countries. Based on the numbers, the country with the least growth is Timor-Leste and the largest is the Lao People's Democratic Republic (Lao PDR). Even so, we can see consistent GDP growth. The colored map shows the position of the external debt-to-GDP ratio in 2021. Based on the ratio, Lao PDR and Cambodia appear to need

to improve external debt management. Even though most countries are in the 20–40% range, these countries still need to keep the ratio safe so that the economy remains stable. Unfortunately, data for Malaysia are not available in the database.

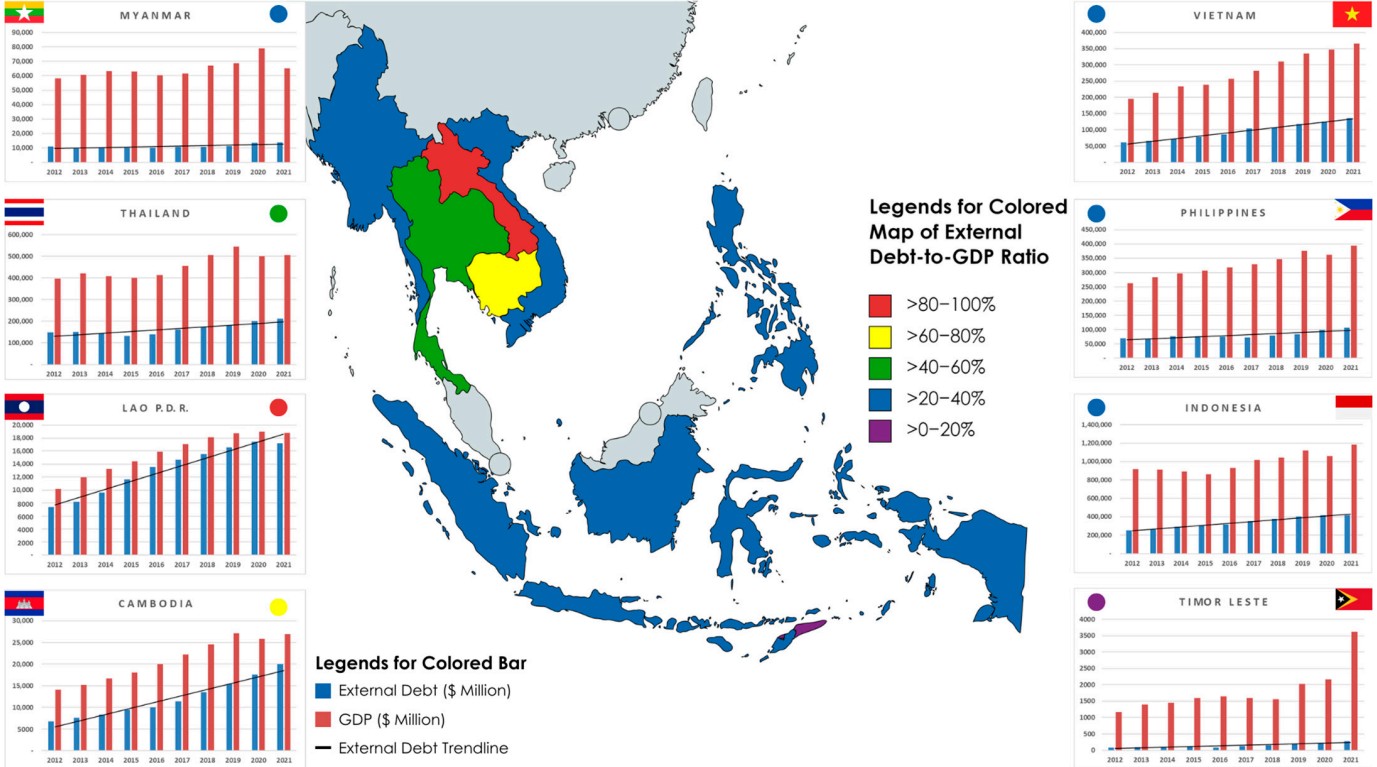

**Figure 1.** Debt, GDP, and ratio of 2021.

External debt growth, like debt growth, is a serious problem (Allen 2013). Greed for easy cash traps the government, causing growth to exceed the debt ceiling. It leads the country to vulnerabilities such as slowing economic growth (Davydenko et al. 2023) or crises (Reinhart and Rogoff 2010). Debt accumulates annually, making it increasingly difficult for the government to escape debt bondage due to various internal and external factors. While expenditures increase, revenues decline, and the fiscal deficit continues to widen, producing a bleak future economy. The debt growth is worrying for the future. Carney et al. (2014) quoted a satire from Herbert Hoover: "Blessed are the young, for they will inherit the national debt". There is concern that the current generation is only creating debt for the next generation.

The statements above show the persistent lessons we learn like the importance of debt management in external financing (Mehran 1986). The five basics of external debt management from Mehran are policy coordination, regulatory environment, operations, accounting, and statistical analysis. Governments, as policymakers, are required to harmonize various policies. They need centralized administration, comprehensive outlining, and supervision. Moreover, they must develop optimized strategies to trade off the costs and benefits obtained from various foreign sources. Therefore, compiling an accurate and comprehensive database will provide initial information for policy determination and an accounting framework for determining external debt.

Our research aims to explore the determinant variable for external debt management in Southeast Asian developing countries. As is generally known, developing countries are vulnerable to slowing economic growth, which is like a middle-income trap. For that reason, we collect literature such as books, news, and articles to find theories, models, methods, empirical results, and rational explanations for the determination of external debt. Then, we arrange the article into the following sections: Abstract, Introduction, Literature

Review, Hypothesis Development, Research Methods, Results, Discussions, Conclusions, Acknowledgements, References, and Appendix.

## 2. Literature Review and Hypothesis Development

### 2.1. Macroeconomics and External Debt

How is debt seen from an economic perspective? This goes back to the two types of views of economists: neoclassical and Keynesian. Phelps (2022) described the neoclassical economist's concern about debt increasing. When the government owes debt, the public reduces its consumption or investment to buy bonds. Afterward, this condition results in a period of slowdown in capital accumulation and productivity. So, governments should keep expenditures based on their revenue to provide a zero or balanced budget. In opposition to the theory, Keynesians see nothing dangerous in increasing debt. We owe debt, from the left to the right, ourselves. Tax revenues generated from economic growth after the government increases its spending and interest income generated from bonds offset concerns about deficit increase. In the short term, macroeconomic variables determine debt.

Wray (2012) identified the three sectoral balances from Godley (1996): domestic private, domestic government, and overseas/foreign balance. The difference between saving and investment (S-I) is the domestic private balance; the difference between tax revenue and expenditure (T-G) is the government balance; and the difference between export and import (X-M) is the overseas/foreign balance. Subsequently, an imbalance in the government sector—a deficit that creates debt—is balanced by a surplus in the other sector. Policies—such as fiscal policy, monetary policy, and trade policy—influence these accounts. Every policy is calculated using socioeconomic indicators. The United Nations (2002) classified various types of indicators into eleven sections, such as (1) production, GDP, capital formation, and saving; (2) labor market; (3) price, wage rate, exchange rate, and interest rate; (4) foreign trade; (5) balance of payments, current account; (6) taxes, government budget, and fiscal deficit; (7) money supply, debits and credits, external and/or government debt, banks, official creditors (govt.), private creditors, balance of payments; (8) business (corporate) income and investment; (9) household income, consumption, saving, and capital formation; (10) social conditions; and (11) environment.

Bittencourt (2015) stated that the external debt determinant is not only based on economic but also political factors. The government's efforts to manage the economy do not optimally materialize without better political institutional arrangements (Abere and Akinbobola 2020). There is no one who is most appropriate when making decisions. However, with good institutional quality, the policies made by the government are more focused and can convince the public. The public needs that trust to for development programs to be successfully implemented by the government. Kaufmann et al. (2004, 2005) provided an example of measuring institutional quality, which is currently also used by the World Bank. The indicators used are the level of perception of (1) voice and accountability; (2) political stability and absence of violence/terrorism; (3) government effectiveness; (4) regulatory quality; (5) rule of law; and (6) control of corruption. They found a link between institutional quality as an indicator of good governance and economic development.

To complement these theories, we collected articles related to the themes above. To gain comprehensive knowledge about the causative factors, we found a few articles focusing on external debt or economic growth. We mapped the authors, publication years, independent variables, methods, number of observations, countries, and periods in Appendix A.

### 2.1.1. Inflation Rate and External Debt

Wray (2012) stated that inflation drives tax revenue growth faster than it drives spending increases. Each positive inflation rate means higher prices of goods and services. When inflation occurs, developing countries with inelastic demand do not have many choices. In the short term, this actually increases tax revenue growth due to increases in prices in the market. Although inflation also increases the value of government expenditures, as

long as tax revenue is greater than the increase in government spending, debt reduction occurs. There are research results that show a negative effect of inflation rate on external debt (Bittencourt 2015; Beyene and Kotosz 2020; Sağdiç and Yildiz 2020; Dawood et al. 2021; Nguyen and Luong 2021; Adane et al. 2018; Okwoche and Nikolaidou 2022). Besides that, there are empirical results that show a positive effect (Waheed 2017).

**Hypothesis 1.** *Inflation rate has a negative effect on external debt.*

### 2.1.2. Exchange Rate and External Debt

Generally, governments denominate financing records and payments in the local currency (Wray 2012). Under conditions of an increase in the exchange rate, the local currency weakens, and the debt account rises. This condition reduces the government's ability to pay debts. The condition can be mitigated by restructuring the debt (Sağdiç and Yildiz 2020). Still, the deficit is wider because of the reduced ability to pay. Likewise, transactions such as imports are carried out by both the government and the private sector. An increase in the exchange rate weakens their ability to pay. There are research results that show a positive effect of exchange rate on external debt (Dawood et al. 2021; Omar and Ibrahim 2021; Adane et al. 2018; Ebiwonjumi et al. 2023; Mijiyawa and Oloufade 2023) while others have a negative effect (Abdullahi et al. 2015; Gokmenoglu and Rafik 2018).

**Hypothesis 2.** *Exchange rate has a positive effect on external debt.*

### 2.1.3. Interest Rate and External Debt

Interest rate is a central bank policy in the monetary sector and is used for things such as regulating money circulation. Goodwin et al. (2015) explained that an increase in interest rate encourages increased saving and decreased investment. Furthermore, Goodwin classified interest rates into nominal interest rates and real interest rates. Real interest rates are the difference between nominal interest rates and inflation. We argue that the public sees an increase in the interest rate as momentum to buy government bonds. Moreover, an interest that exceeds the inflation rate is more valuable. People also reduce borrowing because higher interest rates cause higher burdens on repayment. This surplus between savings and investment provides funds for the government and reduces dependence on external parties. There are research results that show a negative effect of interest rate on external debt (Abdullahi et al. 2015; Waheed 2017; Ebiwonjumi et al. 2023) while others show a positive effect (Brafu-Insaidoo et al. 2019; Mijiyawa and Oloufade 2023).

**Hypothesis 3.** *Interest rate has a negative effect on external debt.*

### 2.1.4. Trade Openness and External Debt

Trade openness is a government policy in the field of foreign trade. Its value is considered to have an ambiguous direction because it is the sum of two opposite directions in accounting, namely, exports and imports (Brafu-Insaidoo et al. 2019; Kızılgöl and İpek 2014). Combes and Saadi-Sedik (2006) explained that trade openness influences the budget deficit through three channels: increased corruption, a change in the supply–demand equilibrium, and a decrease in the government's ability to collect taxes. There are research results that show a positive effect of trade openness on external debt (Dawood et al. 2021; Omar and Ibrahim 2021; Ebiwonjumi et al. 2023; Mijiyawa and Oloufade 2023) while others show a negative effect (Bittencourt 2015; Brafu-Insaidoo et al. 2019; Beyene and Kotosz 2020).

**Hypothesis 4.** *Trade openness has a positive effect on external debt.*

## 2.2. Institutional Quality and External Debt

Institutional quality is a concept regarding the value of an institution in the eyes of the public. Hassoun (2014) argues that there is a clear relationship between good institutional quality and development. Based on theory, we assume that the better a government manages its country, the more its debt can be reduced. The empirical results that we found regarding the relationship between institutional quality and external/public debt still use each indicator as an independent variable (Nguyen et al. 2017; Phuc Canh 2018; Nguyen et al. 2018; Nguyen and Luong 2021; Mehmood et al. 2021). Meanwhile, we did not find anyone who uses it as an index directly for external debt, and the closest relationship is to economic growth (Mensah et al. 2018; Mohd Daud 2020; Samad et al. 2022). We found that external debt has an inverse relationship with economic growth (Mehmood et al. 2021). The positive direction obtained in these studies is estimated to mean that institutional quality has a negative relationship with external debt.

**Hypothesis 5.** *Institutional quality has a positive effect on external debt.*

## 2.3. Macroeconomics, Institutional Quality, and External Debt

Institutions play a key role in shaping a country's economic policies and governance practices (Bruinshoofd 2016). As a policymaker, the government must manage its capabilities and resources to achieve good governance goals and face threats. The government formulates its policy through macroeconomic policy instruments by looking at macroeconomic indicators. We assume that governments with good quality scores can manage their external debts through macroeconomic variables and instruments. From the literature review, we only found the use of institutional quality as a moderator of economic growth. The results of the moderating influence of institutional quality (using each indicator) on the relationship between macroeconomic variables still vary. Some studies stated significance (Nguyen et al. 2017; Phuc Canh 2018) and others did not have significance (Nguyen et al. 2018). Meanwhile, using the institutional quality index, the results still vary greatly for each macroeconomy account (Mohd Daud 2020; Samad et al. 2022). The variety of empirical results shows the importance of this research as a contribution.

**Hypothesis 6.** *Institutional quality has a moderate effect on the relationship between inflation rate and external debt.*

**Hypothesis 7.** *Institutional quality has a moderate effect on the relationship between exchange rate and external debt.*

**Hypothesis 8.** *Institutional quality has a moderate effect on the relationship between interest rate and external debt.*

**Hypothesis 9.** *Institutional quality has a moderate effect on the relationship between trade openness and external debt.*

For ease of understanding, we present the research model here in the form of an image that can be seen in Figure 2.

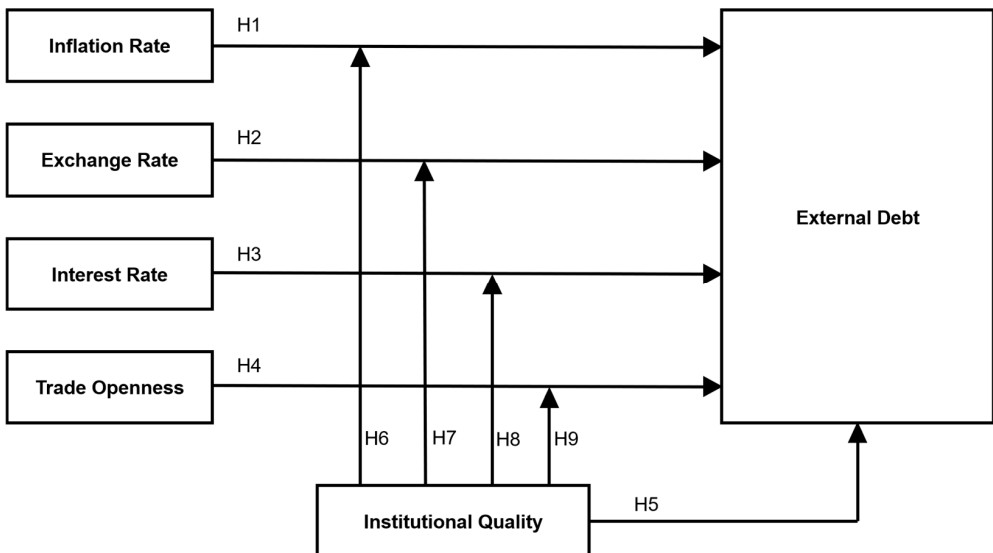

**Figure 2.** Research model.

### 3. Methods

This study uses ordinary least square (OLS) regression and moderated regression analysis (MRA) to examine the determination of external debt. The dependent variables are macroeconomic indicators like inflation rate (INF), exchange rate (EXC), interest rate (INR), and trade openness (TRO). Due to the next theory, we add institutional quality (IQ) as the fifth dependent variable. For the last model, we analyze the interaction between macroeconomic indicators and institutional quality. Our variable composition is based on macroeconomic policies such as monetary and trade policies. Based on our literature review, this composition has not been used before. Likewise, our research object is developing countries in Southeast Asia.

We put together three models for our research objectives, namely:

1.  Relationship of INF, EXC, INR, and TRO on ED

$$ED = \alpha + \beta_1 INF + \beta_2 EXC + \beta_3 INR + \beta_4 TRO + \varepsilon \tag{1}$$

2.  Relationship of INF, EXC, INR, and TRO on ED with the moderation of the institutional quality index (IQ)

$$ED = \alpha + \beta_1 INF + \beta_2 EXC + \beta_3 INR + \beta_4 TRO + \beta_5 IQ + \varepsilon \tag{2}$$

$$ED = \alpha + \beta_1 INF + \beta_2 EXC + \beta_3 INR + \beta_4 TRO + \beta_5 IQ + \beta_6 INF.IQ \\ + \beta_7 EXC.IQ + \beta_8 INR.IQ + \beta_9 TRO.IQ + \varepsilon \tag{3}$$

Data taken from the World Bank's World Development Indicators. The description of each variable can be seen in Appendix B. For data completeness and accuracy in regression analysis, this research used only five of the nine developing countries in Southeast Asia: Indonesia, the Philippines, Thailand, Myanmar, and Timor-Leste. The data taken are from 2008–2019. In addition to purposive sampling, we also eliminated eight lines of extreme data that appeared due to extraordinary events in a particular country and year. The total observations in this research comprise 52 datasets. To see the data used in this research, see Appendix C.

We performed classical assumption tests such as the normality test with one-sample Kolmogorov–Smirnov against unstandardized residues, the multicollinearity test with the variance inflation factor (VIF) approach, and the heteroscedasticity test using the Park test. We skipped the autocorrelation test in regressions with panel data because it was not required. After the classical assumption test, we also analyzed using the determination

test ($R^2$), the F-test, and the *t*-test. Determination tests to obtain model variable accuracy. The F-test and *t*-test were analyzed to examine our research hypotheses for obtaining significance and impact.

## 4. Results

The following is a description of the data we use:

Table 1 describes the variable data used in this study. Variables show homogeneity data like EXD, INR, TRO, and IQ and heterogeneity data like INF and EXC. It is still too early to conduct hypothesis analysis at this stage. However, the data are still worthy of being displayed and made known to the public.

**Table 1.** Data descriptions.

| Variable | Observation | Min. | Max. | Mean | Std. Dev. |
|---|---|---|---|---|---|
| EXD | 52 | 76.04 | 402,106.45 | 112,627.13 | 110,200.02 |
| INF | 52 | −2.65 | 18.15 | 4.00 | 4.05 |
| EXC | 52 | 1.00 | 14,236.94 | 2852.82 | 4903.68 |
| INR | 52 | −3.85 | 19.16 | 5.81 | 4.62 |
| TRO | 52 | 11.86 | 140.44 | 73.05 | 34.00 |
| IQ | 52 | 8.36 | 46.83 | 35.27 | 10.38 |

The results of the classical assumption test are as follows:

Table 2 shows the results we obtained from the normality test on unstandardized residual data, the multicollinearity test with the VIF approach, and the heteroscedasticity test with Park's test. First, the normality test with Kolmogorov–Smirnov analysis requires the probability value to exceed 0.05, while we obtained 0.2 in all models. Second, the VIF value needed to be free from multicollinearity is between 1 and 10. The results we obtained showed that all variables were free from symptoms of multicollinearity, except for the third model, which was not tested based on Disatnik and Sivan (2016) and McClelland et al. (2017). Last, the probability value of homoscedasticity is more than 0.05, and all variables pass the standard.

**Table 2.** Classical test results.

| Test | Variable | First Model | Second Model | Third Model | Results |
|---|---|---|---|---|---|
| Normality | | 0.200 | 0.200 | 0.200 | Data are normally distributed. |
| Multicollinearity | INF | 2.026 | 6.881 | * | Variables have a high degree of freedom from other variables. |
| | EXC | 1.377 | 3.361 | * | |
| | INR | 2.334 | 7.573 | * | |
| | TRO | 2.086 | 2.101 | * | |
| | IQ | | 5.902 | * | |
| | INF.IQ | | | * | |
| | EXC.IQ | | | * | |
| | INR.IQ | | | * | |
| | TRO.IQ | | | * | |
| Heteroscedasticity | INF | 0.253 | 0.874 | 0.375 | Residual variance tends to be constant. |
| | EXC | 0.314 | 0.347 | 0.051 | |
| | INR | 0.054 | 0.228 | 0.711 | |
| | TRO | 0.596 | 0.338 | 0.661 | |
| | IQ | | 0.671 | 0.662 | |
| | INF.IQ | | | 0.427 | |
| | EXC.IQ | | | 0.052 | |
| | INR.IQ | | | 0.994 | |
| | TRO.IQ | | | 0.324 | |

* Model 3 did not test multicollinearity.

Next, we tested the determination the ($R^2$), F-test, and *t*-test.

According to Table 3, the first model reached 95.6% on the determination test. The second and third models reached 95.9% and 96.6% on the determination test, respectively. These three models have a robust determination. Our models almost have the highest determinations after Beyene and Kotosz (2020) of 98.07 percent.

**Table 3.** Analysis results.

| Test Name | Desc/Variable | First Model | Second Model | Third Model |
|---|---|---|---|---|
| Determination ($R^2$) | $R^2$ | 0.959 | 0.963 | 0.972 |
| | Adjust. $R^2$ | 0.956 | 0.959 | 0.966 |
| F-test | F-value | 275.143 *** | 240.789 *** | 162.187 *** |
| *t*-test | Constant | 13,3154.83 *** | 220,048.03 *** | 143,986.45 |
| | INF | −13,137.72 *** | −17,005.02 *** | −4830.55 |
| | EXC | 23.79 *** | 25.83 *** | 14.16 * |
| | INR | −10,733.74 *** | −14,251.86 *** | −7877.59 |
| | TRO | 362.97 ** | 388.32 ** | −831.57 |
| | IQ | | −1662.74 * | −822.53 |
| | INF.IQ | | | −205.73 |
| | EXC.IQ | | | 0.24 |
| | INR.IQ | | | −53.05 |
| | TRO.IQ | | | 34.01 * |

*** $\varrho$-value ≤ 0.001. ** 0.001 < $\varrho$-value ≤ 0.01. * 0.01 < $\varrho$-value ≤ 0.05.

Based on the F-test, the F-value for each model is large if compared to the critical value of F. The probability value also shows a figure of 0.000, and it is significant in all models. This means the model used can measure a significant correlation between independent variables and external debt in all models.

Complementing the F-test, we performed the *t*-test. In the first model, all the independent variables have a significant relationship ($\varrho$-value < 0.01). The significant negative influence on external debt is shown by the inflation rate and interest rate, while the significant positive influence on external debt is shown by the exchange rate and trade openness. The addition of the institutional quality variable does not change the direction or significance of the other independent variables. The results obtained show that institutional quality has a significant negative effect on external debt. We accept the first to fifth hypotheses.

The moderated regression analysis (MRA) on the third model varied in results. Institutional quality may be insignificant in the relationship between the inflation rate, exchange rate, and interest rate on external debt. However, institutional quality is successfully significant in the relationship between trade openness and external debt. The moderating of institutional quality has a positive effect on relationship between trade openness and external debt.

## 5. Discussions

Variable inflation, exchange rate, interest rate, and trade openness theoretically influence external debt. The empirical result we obtained from these variables significantly impact external debt. This significant outcome is also consistent with results from previous studies (Bittencourt 2015; Waheed 2017; Adane et al. 2018; Beyene and Kotosz 2020; Sağdiç and Yildiz 2020; Dawood et al. 2021; Mensah et al. 2017; Nguyen and Luong 2021; Omar and Ibrahim 2021; Okwoche and Nikolaidou 2022). Some studies yielded an insignificant portion or all of the variables (Abdullahi et al. 2015; Gokmenoglu and Rafik 2018; Brafu-Insaidoo et al. 2019; Ebiwonjumi et al. 2023; Mijiyawa and Oloufade 2023).

Inflation rate has a negative effect on external debt. The influence of a negative inflation rate on external debt is supported by previous research (Bittencourt 2015; Adane et al. 2018; Beyene and Kotosz 2020; Sağdiç and Yildiz 2020; Dawood et al. 2021; Nguyen and Luong 2021; Okwoche and Nikolaidou 2022). Different results from Mensah et al. (2017) found an insignificant negative relationship, while (Waheed 2017) obtained a significantly positive effect. Inflation is the phenomenon of a rise in the prices of goods (Elmendorf and

Mankiw 1998). We argue that the selected Southeast Asian country has inelastic market conditions. In the short term, this causes inflation to increase consumption—including spending—in both the private and government sectors. Adane et al. (2018) explained that when there is higher inflation, the government can obtain more taxes, although this has the impact that government spending also becomes greater. Through value-added tax (VAT), the government derives direct benefits from increasing consumption due to inflation. The Basics of Macroeconomic Accounting by Wray (2012) informs us that an interest rate increase makes the resulting deficit smaller or produces a surplus in the fiscal balance. Martin (2009) also stated something similar. The government also increases its ability to pay its debts. The conclusion drawn is that the negative influence between the inflation rate and external debt is caused by an increase in tax revenue, which is more dominant than an increase in spending due to the effect of increasing prices of goods, based on the results obtained.

Exchange rate has a positive effect on external debt. The positive effect of the exchange rate on external debt follows previous research (Adane et al. 2018; Dawood et al. 2021; Mijiyawa and Oloufade 2023). Our result rejects that of Ebiwonjumi et al. (2023), who obtained a positive relationship that was not significant; the results of research by Abdullahi et al. (2015), who obtained a negative effect; and Gokmenoglu and Rafik (2018), who obtained an insignificant negative relationship. Mijiyawa and Oloufade (2023) mentioned the "original sin" related to government debt borrowed in foreign currency. A country that owes money in another country's currency has an increased burden to pay. The government is burdened by the debt service that is due, plus the difference in value that increases due to the weakening of the exchange rate. This can be handled if the government has adequate foreign exchange reserves. However, if it does not have them, it is very clear that the increase in exchange rate value becomes directly proportional to the increase in the debt's value. In addition, there is a possibility that the government borrows to cover its debts due to its inability to pay the increase caused by the weakening in exchange rate. Apart from that, the weakening exchange rate also has an impact on the foreign trade sector. For countries whose import value is greater than export value, a weakening in exchange rate has the impact of increasing the amount of debt because the value that should be paid increases in that country's currency, as stated in the study "Government Debt" by Elmendorf and Mankiw (1998). This can widen the trade deficit, which increases external debt (Dawood et al. 2021). Based on "The Basics of Macroeconomic Accounting" by Wray (2012), this exchange rate increases the expenditure and import accounts. The conclusion can be drawn that the positive influence of exchange rate on foreign debt is caused by a decrease in the ability to pay debt service, the possibility of new debt to cover debt payments, and the impact on import payment abilities.

Interest rate has a negative effect on external debt. Our result is in accordance with previous research (Abdullahi et al. 2015; Waheed 2017). There are also research results from Ebiwonjumi et al. (2023), who obtained an insignificant negative relationship. The results of this study differ from the research of Brafu-Insaidoo et al. (2019) and Mijiyawa and Oloufade (2023), who obtained a positive influence. The interest rate is a form of policy from a country's central bank. Interest rate and exchange rate management are carried out to achieve competitive growth (Ebiwonjumi et al. 2023). Determination of the interest rate is used to reduce the inflation rate (Wray 2012). Apart from that, the interest rate is also a policy for regulating exchange rates. An increase in interest rate encourages an increase in savings and reduces investment. When savings increase and/or investments decrease, private surplus increases. Furthermore, based on "The Basics of Macroeconomic Accounting" by Wray, if net trade is balanced (zero), then a deficit is formed in proportion to the surplus between saving and investment. As long as the supply of savings is available, external debt does not accumulate, and this amount can even be reduced. This opinion is also supported by the theories presented in Elmendorf and Mankiw (1998). This helps the government to shift external debt into domestic debt.

Trade openness has a positive effect on external debt. Our result is in accordance with previous results from Dawood et al. (2021) and Mijiyawa and Oloufade (2023). While Bittencourt (2015), Brafu-Insaidoo et al. (2019), Omar and Ibrahim (2021), and Ebiwonjumi et al. (2023) obtained an insignificant relationship, Beyene and Kotosz (2020) found a significant negative effect. We know that trade openness is the result of the accumulation of export and import values. Meanwhile, according to Wray (2012) in "The Basics of Macroeconomic Accounting", both have different directions for forming the foreign/trade balance. Exports reduce the value of external debt, while imports increase the value of external debt. The government, through its foreign trade policy, regulates the quantity and type of goods entering and leaving. This is implemented for various reasons, such as price setting and availability, diplomacy, security, and so on. Mijiyawa and Oloufade (2023) explained the two-sided effects of trade openness. Regarding fiscal balance, trade openness presents opportunities for the government to obtain greater tax revenues due to greater export and import transactions, resulting in more international trade taxes (international trade tax revenue). Apart from that, trade openness also encourages more government spending due to price increases as a result of increasing global demand and currency exchange rates (Elmendorf and Mankiw 1998). Dawood et al. (2021) argued that the effect on the household side is the creation of job opportunities and increased public consumption, as well as an increase in the country's foreign exchange reserves. On the other hand, external debt increases if imports are greater than exports. Based on the results obtained in this research, the negative relationship between trade openness and external debt is due to the value of imports being greater than that of exports, while on the tax revenue side, it this difference cannot be covered.

Institutional quality has negative effects on external debt. The negative influence of institutional quality on external debt is in accordance with previous research (Nguyen et al. 2017). There are also research results from Nguyen et al. (2018) that show an insignificant negative direction. This research is also different from the results obtained by Phuc Canh (2018), who produced a positive influence. Apart from these three, there are also partial results in different directions (Mehmood et al. 2021; Nguyen and Luong 2021).

Nguyen et al. (2018) argued that improving institutional quality has a strong impact on the effectiveness of fiscal policy in developing countries. Mensah et al. (2018) stated that countries with better institutional quality receive greater benefits from the amount of external debt. This means that to receive the same benefits, the country should owe less (debt efficiency). Mehmood et al. (2021) concluded that countries with low institutional quality drive fiscal deficits higher, weaken economic sustainability, and tend to increase debt. Apart from the fiscal side, we do not obtain any explanation about the impact on private balance and trade balance. We estimate that the private balance, namely, the amount of savings and investment, are both equally affected by the level of institutional quality. The higher the level of institutional quality, the higher the level of public trust in making savings and investing (Samad et al. 2022). Likewise, regarding foreign balance, as the level of institutional quality increases, the amount of exports and imports increase equally. However, this is a limitation for us to explain further.

What was the result of moderation in this research? Institutional quality's interaction with the inflation rate, exchange rate, and interest rate seems insignificant, subsequently rejecting the sixth to eighth hypotheses. Institutional quality weakens but is insignificant in the relationship between inflation rate and interest rate on external debt, while strengthens but is insignificant in the relationship between exchange rate on external debt. From its effect, we can say that institutional quality supports the macroeconomics policy but is insignificant. We suspect the insignificant result is based on the determination of factors that are too high. From our statistics, we found that macroeconomics and institutions result in high determination and significance on external debt. We argue that the selected developing countries of the ASEAN could maximize efforts on external debt management based on macroeconomic instruments and variables. This does not mean ignoring the importance of institutional quality. Based on our empirical results, the interaction between variables is

still not significant. Furthermore, we assume that there are parts of the institutional quality indicator that are worth developing.

We found a moderating effect of institutional quality on the trade openness and external debt relationship, in agreement with the ninth hypothesis, showing a strengthening effect. This is in line with research by Nguyen et al. (2018), who found a negative interaction between institutional quality and trade openness on economic growth. With many results showing that external debt has an inverse relationship with economic growth (Mehmood et al. 2021), this is justification that our results are in line with Nguyen et al. (2018). Institutional quality strengthens the influence of trade openness on external debt. It was previously known that trade openness is a foreign policy in the form of opening or threatening foreign trade. Trade openness is measured by the number of exports and imports in a country. In countries that have greater imports than exports, the external debt debt rises (Wray 2012). When imports are greater than exports, the government generates a deficit directly if the private sector cannot cover it (Elmendorf and Mankiw 1998). Countries with better institutional quality strengthen this effect. Foreign parties feel more secure and confident in transactions with that country because of political stability, effective regulations, and clarity of regulations. Likewise, in this study, the majority of these countries are importers. This also creates a tendency to increase debt.

## 6. Conclusions

This research produced a suitable model according to the level of determination and simultaneous or partial significance of selected Southeast Asian developing countries. The variables used were inflation rate, exchange rate, interest rate, trade openness, and institutional quality, which are appropriate variables to examine for their influence on external debt.

Regarding the research findings of the selected Southeast Asian developing countries, we divided the empirical results into several parts. Inflation rate, interest rate, and institutional quality had negative effects on external debt, while exchange rate and trade openness positively affected external debt. Institutional quality showed insignificant effect on the relationship between inflation rates, exchange rates, and interest rates on external debt. The interaction effect may be weakened (for inflation rate and interest rate) or strengthened (for exchange rate), but they remain insignificant. At the same time, institutional quality significantly strengthens the relationship between trade openness on external debt. We found significant determination factor for external debt management in the selected Southeast Asian developing countries. Even though, based on the results we obtained, not all macroeconomic policies and institutional quality can be interacted with, we still believe that this theory needs to be tested again. Institutional quality moderating the relationship between trade openness and external debt can be seen as evidence for that purpose.

We suggest external debt management in the selected Southeast Asian developing countries focus on macroeconomic policy. For increasing purposes, governments can use policies that increase exchange rates or trade openness (particularly import). For high institutional quality in particular, imports should be prioritized. Furthermore, the government can focus on policies that increase inflation rates and interest rates for decreasing purposes. Moreover, an enhancement in institutional quality may be feasible, considering that this value is still low. Governments also need to be careful about making decisions based on macroeconomic policy. The short-term effect may have different impacts in the long term.

This research has several limitations First, there is potential bias regarding the institutional quality variable, which consists of six sub-variables. Based on these limitations, there are many possibilities for using the institutional quality sub-variable as the following variable. Second, the trade openness variable has two directions. There should be more research on these u-curve possibilities. We suspect there is a safe point for the government to make trading decisions while maintaining debt levels. Third, this research is based on short-term usage. There is a future research opportunity about long-term usage and pairing within.

**Author Contributions:** Conceptualization, E.H., A.K. and N.N.; methodology, E.H.; software, E.H.; validation, E.H., A.K. and N.N.; formal analysis, E.H., A.K. and N.N.; investigation, E.H.; resources, E.H.; data curation, E.H.; writing—original draft preparation, E.H.; writing—review and editing, E.H.; visualization, E.H.; supervision, A.K. and N.N.; project administration, E.H.; funding acquisition, E.H. All authors have read and agreed to the published version of the manuscript.

**Funding:** This research was funded by the Indonesia Endowment Fund for Education (LPDP RI), grant number 202112110408166.

**Informed Consent Statement:** Not applicable.

**Data Availability Statement:** Publicly available datasets were analyzed in this study. These data can be found here: https://databank.worldbank.org/source/world-development-indicators accessed on 9 August 2023.

**Acknowledgments:** We are grateful to the two experienced academics who conducted blind peer reviews for their constructive suggestions. Our thanks are due to all discussants for their time, especially the professors and lecturers of the Master of Accounting Study Program at Hasanuddin University. Lastly, thankful to the Indonesia Endowment Fund for Education.

**Conflicts of Interest:** The authors declare no conflicts of interest. The funders had no role in the design of the study; in the collection, analyses, or interpretation of data; in the writing of the manuscript; or in the decision to publish the results.

## Appendix A. Empiric Literatures

| Authors (Year) | Independent Variables | Model | Obs. | Countries | Period |
|---|---|---|---|---|---|
| Adane et al. (2018) | Inflation, budget deficit, exchange rate | ARDL | 36 | Ethiopia | 1981–2016 |
| Beyene and Kotosz (2020) | Private balance, trade deficit, budget deficit, debt service, trade openness, growth rate of major advanced countries, inflation, growth | ARDL | 36 | Ethiopia | 1981–2016 |
| Bittencourt (2015) | Growth, trade openness, liquid liabilities, inflation, urbanization, executive constraint, government share of GDP, population, inequality | Pooled OLS, one-way and two-way fixed effect, and fixed effect with instrumental variables | 342 | 9 South American countries | 1970–2007 |
| Brafu-Insaidoo et al. (2019) | Financial liberation, interest rate, domestic money supply, trade openness, GDP, relief initiative | ARDL | 43 | Ghana | 1970–2012 |
| Davydenko et al. (2023) | Public debt, debt growth/population growth, debt growth/income growth, household savings/debt, debt/GDP, working population, debt growth to unemployment rate, inflation rate | Descriptive | 8 | Ukraine | 2013–2021 |
| Dawood et al. (2021) | Growth, government expenditure, investment, trade openness, inflation | GMM | 160 | 32 Asian developing and transitioning economies | 1995–2019 |
| Ebiwonjumi et al. (2023) | Internal debt, external debt, real interest rate, exchange rate, trade openness | Multiple linear regression | 141 | Nigeria | 1986–2021 |
| Gokmenoglu and Rafik (2018) | GDP, exchange rate, recurrent expenditure, capital expenditure | Vector error correction model | 44 | Malaysia | 1970–2013 |

| Authors (Year) | Independent Variables | Model | Obs. | Countries | Period |
|---|---|---|---|---|---|
| Mehmood et al. (2021) | Voice and accountability, political stability, government effectiveness, regulatory quality, rule of law, control of corruption | OLS, quantile regression, and robust regression | 22 | Pakistan | 1996–2018 |
| Mensah et al. (2017) | Government consumption expenditure, government investment expenditure, tax revenue, domestic debt, GDP, inflation | Panel vector autoregression | 941–1155 | 24 African countries | 1980–2010 |
| Mijiyawa and Oloufade (2023) | Nominal exchange rate, economic growth, saving–investment gap/GDP, terms of trade, lending interest rate, political right | Fixed effects and GMM | 437–515 | 50 Low- and middle-income countries | 1970–2017 |
| Nguyen and Luong (2021) | Government revenue, public expenditure, inflation, unemployment, voice and accountability, political stability, government effectiveness, regulatory quality, rule of law, control of corruption | OLS, random effects, and two-step GMM | 513 | 27 transition countries | 2000–2018 |
| Okwoche and Nikolaidou (2022) | Presence of conflict, arms imports, military expenditure/GDP, real GDP, fiscal balance/GDP, inflation, oil price | ARDL | 51 | Nigeria | 1970–2020 |
| Omar and Ibrahim (2021) | Exchange rate, export, GDP, government expenditure, domestic investment | ARDL | 39 | Somalia | 1980–2018 |
| Sağdiç and Yildiz (2020) | Growth, public expenditure, inflation | Panel regression | 161 | Azerbaijan, Georgia, Kazakhstan, Kyrgyzstan, Uzbekistan, Tajikistan, Turkmenistan | 1995–2017 |
| Phuc Canh (2018) | Regulatory quality and control of corruption | GMM estimators | 260 | 20 emerging markets | 2002–2014 |
| Nguyen et al. (2018) | Control of corruption, government effectiveness, political stability, regulation quality, rule of law, voice of accountability | GMM estimators | 406 | 29 emerging markets | 2002–2015 |
| Mensah et al. (2018) | Institutional quality index | GMM system | 192–216 | 36 Sub-Saharan African countries | 1996–2013 |
| Mohd Daud (2020) | Institutional quality index | GMM system | 636 | 53 countries | 2005–2016 |
| Nguyen et al. (2017) | Control of corruption, government effectiveness, political stability, regulation quality, rule of law | Fixed effect, random effect | 307–308 | 28 Asia Pacific countries | 2002–2013 |
| Samad et al. (2022) | Institutional quality index | Bias-corrected least square dummy | 228 | 43 nations | 1984–2018 |

## Appendix B. Variable Descriptions

| Indicator | Table Name/Source * | Description | Value |
|---|---|---|---|
| External debt (ED) | External debt stocks, total | Total external debt is debt owed to nonresidents repayable in currency, goods, or services. | USD Million |
| Inflation (INF) | Inflation, GDP deflator (annual %) | Inflation as measured by the annual growth rate of the GDP implicit deflator shows the rate of price change in the economy as a whole. | % |
| Exchange rate (EXC) | Official exchange rate (LCU per USD, period average) | Official exchange rate refers to the exchange rate determined by national authorities or to the rate determined on the legally sanctioned exchange market. | >0 |
| Interest rate (INR) | Real interest rate | Real interest rate is the lending interest rate adjusted for inflation as measured by the GDP deflator. | % |
| Trade openness (TRO) | Trade (% of GDP) | Trade is the sum of exports and imports of goods and services measured as a share of gross domestic product. | % |
| Institutional quality (IQ) | * Author calculation from six indicators below (average) | Average value from institutional quality indicators. | 0–100 |
| Voice and accountability | Voice and accountability: percentile rank | Voice and accountability captures perceptions of the extent to which a country's citizens are able to participate in selecting their government, as well as freedom of expression, freedom of association, and a free media. | 0–100 |
| Political stability and absence of violence | Political stability and absence of violence/terrorism: percentile rank | Political stability and absence of violence/terrorism measures perceptions of the likelihood of political instability and/or politically motivated violence, including terrorism. | 0–100 |
| Government effectiveness | Government effectiveness: percentile rank | Government effectiveness captures perceptions of the quality of public services, the quality of the civil service and the degree of its independence from political pressures, the quality of policy formulation and implementation, and the credibility of the government's commitment to such policies. | 0–100 |
| Regulatory quality | Regulatory quality: percentile rank | Regulatory quality captures perceptions of the ability of the government to formulate and implement sound policies and regulations that permit and promote private sector development. | 0–100 |
| Rule of law | Rule of law: percentile rank | Rule of law captures perceptions of the extent to which agents have confidence in and abide by the rules of society, and in particular the quality of contract enforcement, property rights, the police, and the courts, as well as the likelihood of crime and violence. | 0–100 |
| Control of corruption | Control of corruption: percentile rank | Control of corruption captures perceptions of the extent to which public power is exercised for private gain, including both petty and grand forms of corruption, as well as "capture" of the state by elites and private interests. | 0–100 |

* The data sourced from the author's calculation.

**Appendix C. Sample Collected**

| Economic | Country | Code | Year | | | | | | | | | | | |
|---|---|---|---|---|---|---|---|---|---|---|---|---|---|---|
| | | | 2008 | 2009 | 2010 | 2011 | 2012 | 2013 | 2014 | 2015 | 2016 | 2017 | 2018 | 2019 |
| Developed | Brunei Darussalam | BRN | – | – | – | – | – | – | – | – | – | – | – | – |
| | Singapore | SGP | – | – | – | – | – | – | – | – | – | – | – | – |
| Developing | Indonesia | IDN | ✔ | ✔ | ✔ | ✔ | ✔ | ✔ | ✔ | ✔ | ✔ | ✔ | ✔ | ✔ |
| | Cambodia | KHM | – | – | – | – | – | – | – | – | – | – | – | – |
| | Lao PDR | LAO | – | – | – | – | – | – | – | – | – | – | – | – |
| | Myanmar | MMR | – | – | – | – | ✔ | ✔ | ✔ | ✔ | ✔ | ✔ | ✔ | ✔ |
| | Malaysia | MYS | – | – | – | – | – | – | – | – | – | – | – | – |
| | Philippines | PHL | ✔ | ✔ | ✔ | ✔ | ✔ | ✔ | ✔ | ✔ | ✔ | ✔ | ✔ | ✔ |
| | Thailand | THA | ✔ | ✔ | ✔ | ✔ | ✔ | ✔ | ✔ | ✔ | ✔ | ✔ | ✔ | ✔ |
| | Timor-Leste | TLS | – | – | – | – | ✔ | ✔ | ✔ | ✔ | ✔ | ✔ | ✔ | ✔ |
| | Vietnam | VNM | – | – | – | – | – | – | – | – | – | – | – | – |

✔: data used, −: data left.

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
