# Peer review of "External Debt Determinants: Do Macroeconomic and Institutional Ones Matter for Selected ASEAN Developing Countries?"

_economies, doi:10.3390/economies12010007_

Round 1

Reviewer 1 Report

Comments and Suggestions for Authors

The authors take it as axiomatic that public debt is a problem. In fact, this issue is highly controversial and is under debate. See the following references:

Kelton, Stephanie (2020) The Deficit Myth: Modern Monetary Theory and How to Build a Better Economy (London: John Murray).

Ventura, Jaume and Voth, Hans-Joachim (2015) ‘Debt into Growth: How Sovereign Debt Accelerated the First Industrial Revolution’, NBER Working Paper 21280, National Bureau for Economic Research, Cambridge MA.

Wray, L. Randall (2012) Modern Money Theory: A Primer on Macroeconomics for Sovereign Monetary Systems (London and New York: Palgrave Macmillan). 

The authors should rewrite the sentences where they say or imply that public debt is a problem. The paper can remain neutral on this question. 

Comments on the Quality of English Language

Generally OK. 

Author Response

Dear reviewer,
thank you very much for taking the time to review this manuscript. Please find the detailed responses below and the corresponding revisions/corrections highlighted/in track changes in the re-submitted files.
Here is the point from the reviewer,

The authors take it as axiomatic that public debt is a problem. In fact, this issue is highly controversial and is under debate. See the following references:

Kelton, Stephanie (2020) The Deficit Myth: Modern Monetary Theory and How to Build a Better Economy (London: John Murray).

Ventura, Jaume and Voth, Hans-Joachim (2015) ‘Debt into Growth: How Sovereign Debt Accelerated the First Industrial Revolution’, NBER Working Paper 21280, National Bureau for Economic Research, Cambridge MA.

Wray, L. Randall (2012) Modern Money Theory: A Primer on Macroeconomics for Sovereign Monetary Systems (London and New York: Palgrave Macmillan).

The authors should rewrite the sentences where they say or imply that public debt is a problem. The paper can remain neutral on this question.
------------------------------------------------------------------------------------

Thank you for pointing this out. We agree with this comment. Therefore, we have rewritten the introduction and focused on External Debt. Now, the article has focused on external debt from introductions to conclusions.
We also send rewritten articles.
Red letters are rewritten sentences. On the right there is a comments section which is an additional explanation for what we changed.
------------------------------------------------------------------------------------
About reference, we have used Wray (2012) as our reference book, especially regarding the Basics of Macroeconomic Accounting.
------------------------------------------------------------------------------------Regarding neutrality, we have written the two sides of the economic perspective from Phelps (2022) in the literature review. In the first paragraph of the literature review, we rewrote the summary from Phelps (2022) about neoclassical and Keynesian perspectives. From both points of view, it is clear that our position does not support either one. Furthermore, we focus on external debt management by its determinants.
------------------------------------------------------------------------------------
we hope these changes will make this article suitable for publication.

Our grateful to the team.
Authors.

Reviewer 2 Report

Comments and Suggestions for Authors

External Debt Determinants: Do Macroeconomic and Institutional Matter for Selected ASEAN Developing Countries?

 I have doubts about the correctness and precision of interpreting the theories and the reviewed literature. Many statements are imprecise, forcing the reader to guess what they mean. There is a lack of scrutiny in the paper editing. 

1.      The title of the article needs rewriting. I propose „External Debt Determinants: Do Macroeconomic and Institutional Ones Matter for Selected ASEAN Developing Countries?

2.      I am confused about the kind of debt investigated in the study. What debt is under consideration: either state  external (foreign) debt (the liabilities owed to all non-residents by all residents) or exclusively public (general government)  external (foreign) debt? It should be precisely articulated in the paper’s title and the Introduction. Literature review and empirical research should also be strictly connected with the chosen debt. The definition of external debt will help understand the independent variable explained empirically. For the difference between the two, see, for example, the referenced publication: Bittencourt, Manoel. 2015. “Determinants of Government and External Debt: Evidence from the Young Democracies of South America.”

3.      The empirical part in the Introduction better fits the sections following the literature review. 

4.      The research goals (objectives) are unclear and not straightforward. („The objectives of this research are in line with the objectives of external debt management in ASEAN developing countries” - lines 60-61

5.      I do not understand how the collected literature can "be used to develop (...)  populations"? – lines 61-63

6.      Interpretation of Phelps (2020) as reproduced in the paper (lines 71-73) "When the government owes the debt, the public reduces its consumption to buy the bonds. Afterward, this condition results in a period of slowdown of capital accumulation and productivity”) suggests misunderstanding or excessive shortening of the original thought. It is not lower consumption that reduces private capital and accumulation but savings allocated for Treasury bonds instead of investment loans to the private sector (crowding out effect).

7.      Generally speaking, the Authors should be more careful about the precision of their statements. For instance, "Wray (2012) identifies from (Godley 1996) the three sectoral balances: Domestic Private, Domestic Government, and Overseas/Foreign Balance. Subsequently, the government deficit generates an equal surplus on the other side. As long as net trade was balanced (zero), it resulted in domestic wealth" (lines 90-83). Does the last sentence refer to the balance of payments (in particular, current account - income flows – interest flows), or does it refer to the foreign trade balance, and if so, how is it connected with the public debt? Godley's working paper („Money, Finance and National Income Determination: An Integrated Approach”) deals with „the aggregate financial balances of the three main sectors,” which means - national accounts (C+I+G+ X – M). Current Account Balance, except for (X−M), includes net income abroad (NY) having an impact on „domestic wealth”. The outflow of interest on government bonds abroad with net foreign trade equal to zero will not result in (higher) domestic wealth stock.

8.      The higher the inflation rate, the higher the price of goods and services in a country. Then, it will encourage consumption values both in the private and  government sectors” – lines 113-115. The first sentence is confusing. Each positive inflation rate (also lower than in the previous period) means higher prices of goods and services. The second sentence – lack of proof plus needing to consider the law of demand and price elasticities of demand – higher inflation rate can reduce consumption spending.

9.      There are research results that have a negative effect” (line 116). Effect of what on what?

10.   The weakening of the currency had a direct effect on the government's ability to pay that year [external debt]. In addition, countries with a trade deficit will also experience difficulties paying in local currency. This can also increase the amount of external debt” – lines 123-125. First, the authors do not explain how these two variables (government deficit/debt and trade balance), if so, are related to each other. Why do countries with a trade deficit experience difficulties paying [public debt] in local currency?

11.   Interest Rate is a government policy in the monetary sector” – line 133. This is the competence of the monetary authority (central bank), not the government.

12.  Subsequent hypotheses in the text have justifications relating to variables not included in the formulation of these hypotheses. For instance, “H1. Inflation Rate has a Negative Effect on External Debt” , which is followed by the rationale concerning the exchange rate. This error keeps repeating. I propose putting the hypothesis first and then justifying it. Hypotheses 4 and 5 have the same wording

13.  “Through macroeconomic indicators, the government formulates the policies and rules that it will implement” – lines 173-174. The government formulates its policy through macroeconomic policy tools (instruments) looking at macroeconomic indicators such as unemployment, economic growth, etc.

14.  Data sources for chosen macroeconomic variables and the exact characteristics of these variables (e.g., how exchange rates in the investigated countries are quoted – directly or indirectly and to what currency, what kind of interest rates are taken) are missing in the methodological part. Despite the note in the text that Appendix B contains the data used, it does not include their specification and characteristics.

15.  As we understand, inflation is a condition of increasing prices of goods as stated in the Theory of Government Debt from Elmendorf and Mankiw (1998)” – lines 288-289. I don't understand the meaning of this sentence at all. Inflation is not a condition of increasing prices of goods but is the phenomenon of a rise in these prices. Elmendorf and Mankiw's paper presents the conventional theory of government debt, emphasizing aggregate demand in the short run and crowding out in the long run.

Editorial remarks

1.     Figure 1. – title is not precise

2.     To make figures more readable, they should incorporate legends (instead of describing bar colors in the text, e.g., Figure 1).

3.     Line 102 “used by world banks” – World Bank

4.     Referenced revealed in the text but lacking in the Reference list, e.g. Goodwin et al. (2015), Hassoun (2014)

5.     Instead of using "The table above", one can use "Table 1, Table 2., etc

6.     The empirical result we obtained from these variables significantly on external debt” (lines 276-277) – significantly impact on.

Comments on the Quality of English Language

Economic terms need clarification.

Author Response

Dear reviewer,
thank you very much for taking the time to review this manuscript. Please find the detailed responses below and the corresponding revisions/corrections highlighted/in track changes in the re-submitted files.

Here is our response to the General Evaluation.
We have rewritten many sentences in the whole article section. In the new manuscript, we have focused on external debt from introductions to conclusions. The change was rewritten in red color.
--------------------------------------------------------------------------------------
Here is our response to the point-to-point comments from reviewer.
1.      The title of the article needs rewriting. I propose „External Debt Determinants: Do Macroeconomic and Institutional Ones Matter for Selected ASEAN Developing Countries?

Our response:
We accepted the reviewer's suggestion to add the word "Ones". This makes sentences clearer.
--------------------------------------------------------------------------------------
2. I am confused about the kind of debt investigated in the study. What debt is under consideration: either state external (foreign) debt (the liabilities owed to all non-residents by all residents) or exclusively public (general government) external (foreign) debt? It should be precisely articulated in the paper’s title and the Introduction. Literature review and empirical research should also be strictly connected with the chosen debt. The definition of external debt will help understand the independent variable explained empirically. For the difference between the two, see, for example, the referenced publication: Bittencourt, Manoel. 2015. “Determinants of Government and External Debt: Evidence from the Young Democracies of South America.”

Our response:
In line with First Reviewer, we have rewritten the introduction. Next, we made changes to several sections, so that now everything focuses on external debt.
--------------------------------------------------------------------------------------
3.  The empirical part in the Introduction better fits the sections following the literature review. 

Our response:
Intending to align the empirical data with the focus (external debt), we changed Figure 1 based on external debt and GDP. Previously, we have used the government debt and GDP.
--------------------------------------------------------------------------------------
4. The research goals (objectives) are unclear and not straightforward. („The objectives of this research are in line with the objectives of external debt management in ASEAN developing countries” - lines 60-61.

Our response:
We have rewritten the research goals to be clearer and straightforward.
"Our research aims to explore the determinant variable for external debt management in Southeast Asian developing countries. As is generally known, developing countries are vulnerable to slowing economic growth, which is like a middle-income trap."
--------------------------------------------------------------------------------------
5. I do not understand how the collected literature can "be used to develop (...)  populations"? – lines 61-63

Our response:
We have rewritten the sentences. We were not developing theory or population. Previously, we intended to obtain theories, models, methods, empirical results, and populations that had been used previously from the literature review.
"For that reason, we collect literature such as books, news, and articles to find theories, models, methods, empirical results, and rational explanations for the determination of external debt"
--------------------------------------------------------------------------------------
6.  Interpretation of Phelps (2020) as reproduced in the paper (lines 71-73) "When the government owes the debt, the public reduces its consumption to buy the bonds. Afterward, this condition results in a period of slowdown of capital accumulation and productivity”) suggests misunderstanding or excessive shortening of the original thought. It is not lower consumption that reduces private capital and accumulation but savings allocated for Treasury bonds instead of investment loans to the private sector (crowding out effect).

Our response:
Our sentence is based on Phelps (2022) who summarizes the neoclassical perspective and writes it as follows:
"If over the period of large-scale government borrowing, the public have cut back purchases of consumer goods to buy the government’s sales of bonds, their wealth is thereby increased at the end of this period, though the nation’s capital stock is not – not appreciably, at any rate. If instead the public have cut back purchases of new issues of company shares to buy the government’s sales of bonds, the nation’s capital is thereby decreased, though wealth is not." (in the page 2 of 3)
So, we added about reducing investment and kept about reducing consumption.
--------------------------------------------------------------------------------------
7. Generally speaking, the Authors should be more careful about the precision of their statements. For instance, "Wray (2012) identifies from (Godley 1996) the three sectoral balances: Domestic Private, Domestic Government, and Overseas/Foreign Balance. Subsequently, more government deficit generates an equal surplus on the other side. As long as net trade was balanced (zero), it resulted in domestic wealth" (lines 90-83). Does the last sentence refer to the balance of payments (in particular, current account - income flows – interest flows), or does it refer to the foreign trade balance, and if so, how is it connected with the public debt? Godley's working paper („Money, Finance and National Income Determination: An Integrated Approach”) deals with „the aggregate financial balances of the three main sectors,” which means - national accounts (C+I+G+ X – M). Current Account Balance, except for (X−M), includes net income abroad (NY) having an impact on „domestic wealth”. The outflow of interest on government bonds abroad with net foreign trade equal to zero will not result in (higher) domestic wealth stock.

Our response:
As in the previous paragraph based on the summary by Phelps, we believe there is a section about increasing wealth. However, we choose to reconstruct this paragraph to focus only on the three sectoral balance constructions mentioned by Wray (2012:37). (S-I)+(T-G)+(X-M)=0
--------------------------------------------------------------------------------------
8.  “The higher the inflation rate, the higher the price of goods and services in a country. Then, it will encourage consumption values both in the private and government sectors” – lines 113-115. The first sentence is confusing. Each positive inflation rate (also lower than in the previous period) means higher prices of goods and services. The second sentence – lack of proof plus needing to consider the law of demand and price elasticities of demand – higher inflation rate can reduce consumption spending.

Our response:
We have rewritten the sentences. The inelastic demand fits well with the developing countries that are the focus of our research. For example, basic consumption such as rice cannot be replaced by any other type of food in Indonesia. Likewise, countries in Southeast Asian are consumers of rice. Here the rewritten sentences:
"Each positive inflation rate means higher prices of goods and services. When inflation occurs, developing countries with inelastic demand do not have many choices. In the short term, this actually increases tax revenue growth due to increases in prices in the market. Although inflation will also increase the value of government expenditures, as long as tax revenue is greater than the increase in government spending.
--------------------------------------------------------------------------------------
9.  “There are research results that have a negative effect” (line 116). Effect of what on what? 

Our response:
We have rewritten the sentences, and sentences like this in other sections.
"There are research results that show a negative effect of inflation rate on external debt."
--------------------------------------------------------------------------------------
10.  “The weakening of the currency had a direct effect on the government's ability to pay that year [external debt]. In addition, countries with a trade deficit will also experience difficulties paying in local currency. This can also increase the amount of external debt” – lines 123-125. First, the authors do not explain how these two variables (government deficit/debt and trade balance), if so, are related to each other. Why do countries with a trade deficit experience difficulties paying [public debt] in local currency?

Our response:
We have rewritten the sentence to better explain the relationship between exchange rate and external debt. Here the rewritten sentences:
"Generally, governments denominate financing records and payments in local currency (Wray 2012). Under conditions of an increase in the exchange rate, the recorded deficit will widen because the government's ability to pay falls. This may be mitigated by restructuring (SaÄŸdiç and Yildiz 2020). But still, the deficit will basically get bigger because of the reduced ability to pay. Likewise, transactions such as imports are carried out by both the government and the private sector. An increase in the exchange rate weakens their ability to pay. There are research results that show a positive effect of exchange rate on external debt."
--------------------------------------------------------------------------------------
11. “Interest Rate is a government policy in the monetary sector” – line 133. This is the competence of the monetary authority (central bank), not the government.

Our response:
We have accepted the correction that interest rate is central bank policy. Here the rewritten sentences:
"Interest rate is a central bank policy in the monetary sector and is used for things such as regulating money circulation"
--------------------------------------------------------------------------------------
12.  Subsequent hypotheses in the text have justifications relating to variables not included in the formulation of these hypotheses. For instance, “H1. Inflation Rate has a Negative Effect on External Debt” , which is followed by the rationale concerning the exchange rate. This error keeps repeating. I propose putting the hypothesis first and then justifying it. Hypotheses 4 and 5 have the same wording

Our response:
Previously, we were written the hypothesis on the last sentences of paragraph as a conclusions. Now, we add a subsubsection in the macroeconomics relationship. (2.1.1. to 2.1.4.).
For the fourth hypothesis, we have rewritten the sentences.
--------------------------------------------------------------------------------------
13. “Through macroeconomic indicators, the government formulates the policies and rules that it will implement” – lines 173-174. The government formulates its policy through macroeconomic policy tools (instruments) looking at macroeconomic indicators such as unemployment, economic growth, etc.

Our response:
We have accepted the correction and rewritten the sentences. Here the rewritten sentences:
"The government formulates its policy through macroeconomic policy instruments by looking at macroeconomic indicators. We assume that governments with good quality scores can manage their external debts through macroeconomic variables and instruments."
--------------------------------------------------------------------------------------
14. Data sources for chosen macroeconomic variables and the exact characteristics of these variables (e.g., how exchange rates in the investigated countries are quoted – directly or indirectly and to what currency, what kind of interest rates are taken) are missing in the methodological part. Despite the note in the text that Appendix B contains the data used, it does not include their specification and characteristics.

Our response:
We have accepted the suggestion and added the new Appendix B (Variable descriptions). The have changed the old Appendix B into Appendix C.
--------------------------------------------------------------------------------------
15. “As we understand, inflation is a condition of increasing prices of goods as stated in the Theory of Government Debt from Elmendorf and Mankiw (1998)” – lines 288-289. I don't understand the meaning of this sentence at all. Inflation is not a condition of increasing prices of goods but is the phenomenon of a rise in these prices. Elmendorf and Mankiw's paper presents the conventional theory of government debt, emphasizing aggregate demand in the short run and crowding out in the long run.

Our response:
We have accepted the correction about phenomenon. We have rewritten the explanation. Here are the rewritten sentences:
Inflation is the phenomenon of a rise in the prices of goods, as stated in the Theory of Government Debt by Elmendorf and Mankiw (1998). We argued that this Southeast Asian country has inelastic market conditions. In the short-term, this causes inflation to increase consumption—including spending—in both the private and government sectors.
--------------------------------------------------------------------------------------
Editorial remarks
1. Figure 1. – title is not precise. 
Our response: We have changed the figure and the title.

2. To make figures more readable, they should incorporate legends (instead of describing bar colors in the text, e.g., Figure 1). 
Our response: We have remade the figure and added the legends in the figure.

3. Line 102 “used by world banks” – World Bank.
Our response: We have accepted the correction about capital letter on "World Bank".

4. Referenced revealed in the text but lacking in the Reference list, e.g. Goodwin et al. (2015), Hassoun (2014).
Our response: We have added the reference list and link.

5. Instead of using "The table above", one can use "Table 1, Table 2., etc
Our response: We have accepted the correction and rewritten the word.

6. “The empirical result we obtained from these variables significantly on external debt” (lines 276-277) – significantly impact on.
Our response: We have accepted the correction.
--------------------------------------------------------------------------------------

It was our pleasure to have been reviewed. The reviewer's comments are like a discussion to improve our knowledge. We hope our response can be accepted by reviewers and editors.

Sincerely,
Authors.

Round 2

Reviewer 1 Report

Comments and Suggestions for Authors

The paper still assumes that the reduction of national debt is necessarily a problem and a priority. For example, the opening sentence of the abstract reads: "Developing countries face the challenge of managing their national debt to evade the bleak economic future." Such statements are highly controversial within economics. First, the controversy should be briefly acknowledged. (Read J M Keynes and modern monetary theory.) Second, commitment to an entirely negative view of public debt is not necessary in this paper. The authors may focus on how public debt may be reduced, without making normative evaluations of whether or not it is essential or a priority. 

Comments on the Quality of English Language

Generally OK. Could be improved in parts. 

Author Response

Dear reviewer,
thank you very much for taking the time to review this manuscript.

Reviewer comment:
The paper still assumes that the reduction of national debt is necessarily a problem and a priority. For example, the opening sentence of the abstract reads: "Developing countries face the challenge of managing their national debt to evade the bleak economic future." Such statements are highly controversial within economics. First, the controversy should be briefly acknowledged. (Read J M Keynes and modern monetary theory.) Second, commitment to an entirely negative view of public debt is not necessary in this paper. The authors may focus on how public debt may be reduced, without making normative evaluations of whether or not it is essential or a priority.

Our response:

We agree to be neutral in the article, as in our conclusion we suggest using policies that reduce or increase the amount of external debt. Because we realize that the government has dynamic needs.
Here are our rewrite abstracts.

Developing nations have the task of effectively managing their external debt. The government is urged to comprehend the decisive component in managing its external debt, despite the varying viewpoints among economists. In addition, the world sees the need for institutional quality to optimize its economic policy. Institutional quality shows accountability, stability, effectiveness, quality, law, and trust. Our research examines the determinant factors of external debt and discusses the policy to manage external debt. We regress the inflation rate, exchange rate, interest rate, trade openness, and institutional quality on external debt. This study also uses moderated regression analysis to examine the interaction between institutional quality and macroeconomic indicators on external debt. We selected 52 samples from five ASEAN developing countries from 2008 to 2019. The first study found that the inflation rate, interest rate and institutional quality have a negative impact on external debt, while the exchange rate and trade openness have a positive impact on external debt. Next, we were surprised that institutional quality could not moderate the relationship between the inflation rate, exchange rate, and interest rate on external debt. Further, it only moderated the relationship between trade openness and external debt. In the end, we discuss the external debt determinants from the selected ASEAN developing countries with the theories.

Your help and guidance mean a lot to me. Thank you for your time and attention.

Sincerely,
authors.

Reviewer 2 Report

Comments and Suggestions for Authors

Generally, the authors considered my suggestions and comments and improved/supplemented the previous article's version. Their responses are satisfactory. However, it would be good if they read the text carefully again and clarify the sentences.

I will only quote a few statements that are too general, which in their current form may be understood in various ways.

Abstract: “The first study results show that the inflation rate and interest rate significantly decrease the external debt, while exchange rate and trade openness significantly increase the external debt. The next study reveals that institutional quality significantly decreases the external debt”.

I advise the authors to be more precise in formulating their sentences. Do the authors want to report a positive (negative) relationship between the variables? The statements should be unambiguous: e.g., An increase (decrease?) in the inflation rate significantly decreases the external debt, etc.

Another example of lack of precision is “An increase in the exchange rate causes an increase of the external debt. When a country's exchange rate weakens against the USD, the value of this variable will rise”. Agree under the condition that  one adds „in local currency” (external debt expressed in local currency).

An added description of the variables (my previous suggestions) - (Appendix B) is helpful but still non-exhaustive. Additionally, it reveals that the empirical study used the real interest rate, i.e., a nominal interest rate corrected for a measure of inflation (unfortunately, it has not yet been determined whether it is the central bank rate, the bond rate, the short-term rate or long-term, etc.). It is worth rethinking if the applied real interest rate is consistent with the description of the relationship between the interest rate and public debt contained, among others, in section 2.1.3.  Moreover, section 2.1.3. is still mostly concerned with public debt, although the Authors have refocused their research on external debt.

Please, the Authors, consider my previous comments #10 and #15 again”.

 The reviewer, myself, does not require a response to the current comments. Suggestions are because the article in its final version should not raise any doubts among readers.

Comments on the Quality of English Language

Minor editing is required.

Author Response

Dear reviewer, thank you very much for taking the time to review this manuscript.

Comments 1: Abstract: “The first study results show that the inflation rate and interest rate significantly decrease the external debt, while exchange rate and trade openness significantly increase the external debt. The next study reveals that institutional quality significantly decreases the external debt”.

I advise the authors to be more precise in formulating their sentences. Do the authors want to report a positive (negative) relationship between the variables? The statements should be unambiguous: e.g., An increase (decrease?) in the inflation rate significantly decreases the external debt, etc.

Response 1:
We agree to rewrite the abstract.
In the abstract, we show the model with the following sentences:
“We regress the inflation rate, exchange rate, interest rate, trade openness, and institutional quality on external debt.”

We agree to use positive/negative instead of increase/decrease. But based on our previous sentences, we write the results in the abstract (lines 13-17) like this:

The first study found that the inflation rate, interest rate, and institutional quality have a negative impact on external debt, while the exchange rate and trade openness have a positive impact on external debt. Next, we were surprised that institutional quality could not moderate the relationship between the inflation rate, exchange rate, and interest rate on external debt. Further, it only moderated the relationship between trade openness and external debt.

Based on our previous sentences, we think it would be clearer to use the variable as a subject than the result (positive/negative) in the relationship.

We rewrite the conclusion for the same reason in the second paragraph (lines 443-448):
“The research findings in the selected Southeast Asia developing countries divide empirical results into several parts. Inflation rate, interest rate, and institutional quality had negative effects on external debt, while exchange rate and trade openness positively affected external debt. Institutional quality has insignificantly interaction affected the relationship between inflation rates, exchange rates, and interest rates on external debt. The interaction effect may be weakened (for inflation rate and interest rate) or strengthened (for exchange rate), but they remain insignificant. At the same time, institutional quality significantly strengthens the relationship between trade openness on external debt. …”

Comments 2:
Another example of lack of precision is “An increase in the exchange rate causes an increase of the external debt. When a country's exchange rate weakens against the USD, the value of this variable will rise”. Agree under the condition that one adds „in local currency” (external debt expressed in local currency).

Response 2:
We delete the sentence that you quote because this sentence still seems to need additional explanation. Even though the explanation is in the middle to the end of the discussion paragraph.

To make it clearer, we reconstructed the paragraph in the development hypothesis section (lines 134-136).

“Generally, governments denominate financing records and payments in the local currency (Wray 2012). Under conditions of an increase in the exchange rate, the local currency weakens, and the debt account rises. This condition reduces the government's ability to pay debts. The condition may be mitigated by restructuring the debt (SaÄŸdiç and Yildiz 2020). Still, the deficit is wider because of the reduced ability to pay. Likewise, transactions such as imports are carried out by both the government and the private sector. An increase in the exchange rate weakens their ability to pay. There are research results that show a positive effect of exchange rate on external debt (Dawood et al. 2021; Omar and Ibrahim 2021; Adane et al. 2018; Ebiwonjumi et al. 2023; Mijiyawa and Oloufade 2023) while others have a negative effect (Abdullahi et al. 2015; Gokmenoglu and Rafik 2018).”

A positive exchange rate means a weakening condition in local currency. This condition reduces the ability to pay debts, in the fiscal or trade sector (import).
​Meanwhile, in the discussion section, we order as follows:

Hypothesis decision, previous empirical comparison, explanation based on theory, and author's opinion (optional).

Comments 3:
“An added description of the variables (my previous suggestions) - (Appendix B) is helpful but still non-exhaustive. Additionally, it reveals that the empirical study used the real interest rate, i.e., a nominal interest rate corrected for a measure of inflation (unfortunately, it has not yet been determined whether it is the central bank rate, the bond rate, the short-term rate or long-term, etc.). It is worth rethinking if the applied real interest rate is consistent with the description of the relationship between the interest rate and public debt contained, among others, in section 2.1.3.  Moreover, section 2.1.3. is still mostly concerned with public debt, although the Authors have refocused their research on external debt.

Please, the Authors, consider my previous comments #10 and #15 again”.

The reviewer, myself, does not require a response to the current comments. Suggestions are because the article in its final version should not raise any doubts among readers.”

Response 3:
We agree that there is a section missing in 2.1.2. namely an explanation of interest rates and their selection. So, we rewrite the paragraph (lines 147-152):

“Interest rate is a central bank policy in the monetary sector and is used for things such as regulating money circulation. Goodwin et al. (2015) explain that an increase in the interest rate will encourage increased saving and decreased investment. Furthermore, Goodwin classifies interest rates into nominal interest rates and real interest rates. Real interest rates are the difference between nominal interest rates and inflation. We argued that the public sees an increase in the interest rate as momentum to buy government bonds. Moreover, interest that exceeds the inflation rate will be more valuable. People also reduce borrowing because higher interest rates cause higher burdens they have to pay. This surplus between savings and investment provides funds for the government and re-duces dependence on external parties. There are research results that show a negative ef-fect of interest rate on external debt (Abdullahi et al. 2015; Waheed 2017; Ebiwonjumi et al. 2023) while others have a positive effect (Brafu-Insaidoo et al. 2019; Mijiyawa and Oloufade 2023).”

In Appendix B, we also add a description according to the World Bank.

We hope that our explanation can be approved by reviewers. Really, we enjoyed discussing with our reviewers.
